# Nonstimulatory peptide–MHC enhances human T-cell antigen-specific responses by amplifying proximal TCR signaling

Xiang Zhao[1], Shvetha Sankaran[1,2], Jiawei Yap[1], Chien Tei Too[1,2], Zi Zong Ho[3], Garry Dolton[4], Mateusz Legut[4], Ee Chee Ren[5], Andrew K. Sewell [4,6], Antonio Bertoletti[3], Paul A. MacAry[1,2,7], Joanna Brzostek [1] & Nicholas R.J. Gascoigne [1,2,7]

Foreign antigens are presented by antigen-presenting cells in the presence of abundant endogenous peptides that are nonstimulatory to the T cell. In mouse T cells, endogenous, nonstimulatory peptides have been shown to enhance responses to specific peptide antigens, a phenomenon termed coagonism. However, whether coagonism also occurs in human T cells is unclear, and the molecular mechanism of coagonism is still under debate since CD4 and CD8 coagonism requires different interactions. Here we show that the nonstimulatory, HIV-derived peptide GAG enhances a specific human cytotoxic T lymphocyte response to HBV-derived epitopes presented by HLA-A*02:01. Coagonism in human T cells requires the CD8 coreceptor, but not T-cell receptor (TCR) binding to the nonstimulatory peptide–MHC. Coagonists enhance the phosphorylation and recruitment of several molecules involved in the TCR-proximal signaling pathway, suggesting that coagonists promote T-cell responses to antigenic pMHC by amplifying TCR-proximal signaling.

[1] Department of Microbiology and Immunology, Yong Loo Lin School of Medicine, National University of Singapore, 5 Science Drive 2, Singapore 117545, Singapore. [2] Immunology Programme, Life Sciences Institute, National University of Singapore, 28 Medical Drive, Centre for Life Sciences, Level 3, Singapore 117456, Singapore. [3] Emerging Infectious Diseases Program, Duke-NUS Graduate Medical School, 8 College Road, Singapore 169857, Singapore. [4] Division of Infection and Immunity, Cardiff University School of Medicine, Henry Wellcome Building, University Hospital Wales, Heath Park, Cardiff CF14 4XN, United Kingdom. [5] Singapore Immunology Network, A*STAR, 8A Biomedical Grove, Immunos #03-06, Singapore 138648, Singapore. [6] Systems Immunity Research Institute, Cardiff University, Tenovus Building, Cardiff CF14 4XN, United Kingdom. [7] NUS Graduate School for Integrative Sciences and Engineering (NGS), National University of Singapore, Centre for Life Sciences (CeLS), #05-01, 28 Medical Drive, Singapore 117456, Singapore. Correspondence and requests for materials should be addressed to J.B. (email: micjmb@nus.edu.sg) or to N.R.J.G. (email: micnrjg@nus.edu.sg)

During the activation of T cells, a small number of foreign antigenic peptides bound to MHC proteins (pMHC) are typically presented in the midst of a vast majority of pMHC-presenting peptides derived from endogenous proteins. T-cell receptor (TCR) binds to pMHC complex with an affinity dependent on the peptide sequence that is presented, whereas the CD8 or CD4 coreceptors can bind to pMHC with affinities independent of the peptide sequence. A T cell needs to identify the limited number of its specific antigenic pMHC among the excess of self pMHC. T cells are very sensitive to antigenic pMHC and can be activated by a single-antigenic pMHC[1], yet, at the same time they require cross-linking of TCRs in order to be stimulated[2]. Understanding how T cells identify and differentiate the small pool of antigenic pMHC molecules from the endogenous pMHC molecules, and the role of endogenous pMHC during the T-cell response to specific antigenic pMHC can provide critical insights into early molecular events during T-cell activation. Several studies demonstrated that simultaneous presentation of nonstimulatory pMHC in the presence of antigenic pMHC can significantly enhance mouse T-cell responses to antigenic pMHC[3–5]. This phenomenon is termed coagonism[2,6,7]. A heterodimer of antigenic pMHC with certain nonstimulatory pMHC, but not monomers of antigenic pMHC, can enhance mouse CD4+ T-cell responses[3]. However, it was unclear why this coagonist activity did not work for all kinds of nonstimulatory pMHC molecules. Coagonism has also been demonstrated in mouse OT-I CD8+ T cells, but it had no requirements for specific sequences of the coagonist peptides[4,5]. Hence, there is clear evidence for a role for the large excess of endogenous nonstimulatory pMHC complexes in antigen-specific mouse T-cell activation; however, the molecular mechanism underlying this effect is unknown

The molecular interactions required for coagonism initially appeared to differ between MHC class I (MHCI)-restricted CD8+ T cells and MHC class II (MHCII)-restricted CD4+ T cells. The fact that not all of the tested nonstimulatory peptides could induce coagonism in CD4+ T cells[3], while they could in OT-I CD8+ T cells[4,5] was a conundrum. These apparent differences were resolved by demonstrating that the requirement for specific peptides as coagonists depends on the particular TCR system, primarily on the strength of the coreceptor interaction with the pMHC[8]. While binding of CD4 to nonstimulatory pMHCII was not necessary for the coagonism to be effective[3], binding of CD8 to nonstimulatory pMHCI was absolutely essential for coagonism[8]. If this interaction was strong (e.g., with H2-Kb, as in the OT-I TCR system), then there was no measurable requirement for the TCR to recognize the self-peptide in the coagonist MHC molecule. On the other hand, if the CD8 interaction with the coagonist was weaker (e.g., with H2-Db, recognized by F5 TCR), then the TCR required the interaction with the coagonist and could distinguish between different nonstimulatory coagonist pMHC[8]. The weak interaction between CD4 and MHCII[9], therefore explains the peptide specificity of coagonism in CD4+ T cells[3]. Human CD8–MHCI interactions extend over a wide range of binding affinities and are mostly weaker than those between mouse CD8 and H2-Kb[10], suggesting that the molecular requirements for coagonism during human T-cell recognition may differ from those in murine T cells. Critically, it is not known how the presence of coagonist pMHC complexes influences downstream TCR signaling pathways. Furthermore, previous research has mostly focused on mouse T-cell responses with limited examination of coagonism during human T-cell activation.

Coagonism has important implications for human immune responses. Expression of cell surface HLA-C was shown to be associated with the cytotoxic response to HIV infection; high

HLA-C expression was also shown to increase the risk of Crohn's disease[11]. In many viral infections, viruses downregulate cell surface MHC molecules using various strategies[12]. Moreover, several solid and hematopoietic tumors downregulate MHC expression[13]. However, it is not known if this reduction in total MHC affects human T-cell responses due to a reduction in antigenic pMHC amount or to a coagonist-mediated mechanism.

Very little is currently known about coagonism during human T-cell activation, primarily due to the genetic and experimental constraints of working with human systems. In order to work with an antigen-specific TCR, only well-described cytotoxic T lymphocyte (CTL) clones are readily available. There is some evidence for coagonism in human CD8+ T-cell activation, as nonstimulatory pMHC were found to provide a cooperative contribution to the human CTL response to antigenic pMHC presented on carrier quantum dots[14]. However, this study was based on an artificial and nonphysiological antigen-presenting system instead of investigating direct interactions between antigen-presenting cells (APCs) and human T cells. Moreover, the molecular interactions and the mechanism of coagonism during human T-cell activation are currently unknown. To fill in this gap, we decided to use a cell-based antigen presentation system to present antigenic and nonstimulatory pMHC to stimulate human CTL clones specific for an epitope of the Hepatitis B virus (HBV).

Here, we find that nonstimulatory peptides enhance HBV antigen-specific human CD8+ T-cell responses to small amounts of HBV-derived antigen. CD8 binding, but not TCR binding, to the nonstimulatory pMHCI is required for coagonism to help T-cell activation. Nonstimulatory pMHCI also enhances the T-cell response when CD8 binding to the MHCI molecule presenting the antigenic peptide is abrogated. Mechanistically, coagonist pMHC enhances the phosphorylation of CD3ζ, Zap70, LAT, PLC-γ1, and Erk, and recruits more phosphorylated CD3ζ, phosphorylated Lck, and phosphorylated Zap70 into the immunological synapse, demonstrating that coagonist pMHC amplifies proximal TCR signaling strength during recognition of a limited amount of antigen. Unexpectedly, this coagonsim-induced increase in CD3ζ phosphorylation is not accompanied by an enhanced TCR/CD3 complex cell surface downregulation, suggesting that coagonism does not increase the number of triggered TCR molecules (bystander activation), but acts to amplify signal from TCRs engaged by agonist pMHC. This study confirms a role for coagonism in human CD8+ T-cell activation using physiological antigen-presenting cells, and shows that coagonism acts at the level of the TCR-proximal signaling pathway. It also demonstrates that the coreceptor can work by binding to a different pMHC molecule than the one presenting antigen to the TCR.

## Results

**T-REx CHO cells expressing single-chain pMHC trimer as APC.** We previously used the mouse *Tap2*-deficient cell line RMA-S as an APC for mouse coagonist studies[4,5,8]. A human *Tap2*-deficient cell line, T2, was therefore tested as an APC for human coagonist studies. However, we found that T2 cells already express a relatively high amount of HLA-A*02:01 (HLA-A2) on the surface, and that pulsing with peptides did not significantly upregulate the surface amount of HLA-A2 (Supplementary Fig. 1). HLA-A2 on T2 cells present signal sequence-derived peptides, which may explain this result[15], leading us to look for another model system. In order to express human peptide–MHC complexes in a xenogenic cell line (no human MHC expression), a tetracycline-regulated expression (T-REx) CHO cell line was used[8]. T-REx cells express the tet repressor

under blasticidin selection. We transfected T-REx cells with recombinant vectors for human single-chain peptide–MHCI complex (sc pMHCI). Such single-chain trimers express pMHCI as a polypeptide by linking the peptide, β2-microglobulin and MHC heavy chain with flexible linkers[8,16,17]. A tetracycline-inducible vector pcDNA5/TO was used to express the HIV GAG peptide (SLYNTVATL) or two different HBV-derived peptides, E183-91(FLLTRILTI: "E183") and C18-27 (FLPSDFFPSV: "C18"), as a sc trimer of A2 in T-REx cells. Without doxycycline, the transfected T-REx cells express small amounts of "leaky" (low amount) sc pMHCI. Strong expression of (high amount) sc pMHCI is induced by adding 50 ng ml⁻¹ doxycycline. The expression of cell surface HLA-A2 was detected by anti-HLA-A2 antibody staining (Fig. 1a). "TCR-like" antibodies that recognize HLA-A2 in association with the HBV peptides[18] described above were used to quantitate the expression of the specific epitopes. Anti-C18-A2 or anti-E183-A2 antibodies showed similar expression profiles confirming the correct peptide–MHC expression (Fig. 1b). However, some cross-recognition was also observed with the TCR-like antibodies, wherein high amount of E183-HLA-A2 could be partially recognized by anti-C18-A2 antibody, and high amount of C18-HLA-A2 could be partially recognized by anti-E183-A2 (Fig. 1c).

In order to confirm that HLA in the single-chain format can be recognized by the cognate TCR, a human CTL line bearing TCR specific for E183-HLA-A2 complex[18] was cocultured with the stably transfected T-REx cell panel for 3 h, and the expression of the cytokines TNF and IFN-γ, and the degranulation marker CD107a were detected. The data showed that sc pMHCI induced CD8⁺ T-cell effector functions in an antigen-specific manner (Fig. 1d).

**Nonstimulatory pMHC enhance human CTL response to antigen.** To investigate coagonism in human E183 CTL, we cloned sc GAG-HLA-A2 into the constitutive expression vector pcDNA3-Clover, which was supertransfected into a T-REx clone that could simultaneously express low amount of sc E183-HLA-A2. Single cell sorting was performed after supertransfection and antibiotic selection. Total surface expression of HLA-A2 was measured by staining with anti-HLA-A2 (Fig. 2a). The amount of E183-HLA-A2 under repressed conditions was the same in the presence or absence of supertransfected GAG-HLA-A2 as determined using E183-A2 antibody staining (Fig. 2b). In order to test if coagonism occurs during human T-cell activation, the panel of T-REx clones was cocultured with human E183-specific CTL for 3 h, following which, cell surface expression of the

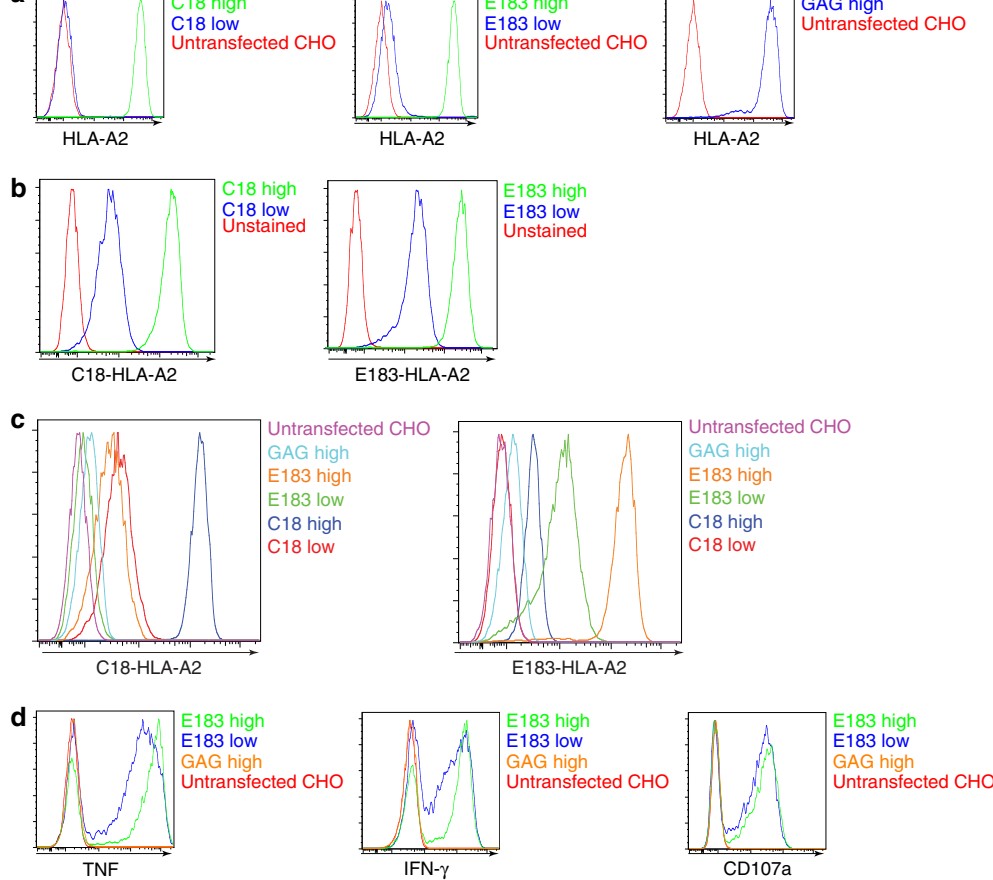

**Fig. 1** Inducible expression of human single-chain (sc)-MHCI in T-REx CHO cells and their immunogenicity to human CTL. **a** anti-HLA-A2 staining of T-REx CHO clones expressing low amount or high amounts of (doxycycline-induced) human single-chain (sc)-MHCI. **b** TCR-like antibody anti-C18-HLA-A2 or anti-E183-HLA-A2 staining of T-REx CHO clones expressing low or high concentrations of C18/E183-HLA-A2 sc pMHC. **c** TCR-like antibody anti-C18-HLA-A2 or anti-E183-HLA-A2 staining of T-REx CHO clones expressing low or high C18/E183-HLA-A2 sc pMHC, high concentration of GAG-HLA-A2 sc pMHC, and unstransfected T-REx CHO cells. **d** Human E183 epitope-specific CTL were cultured with the indicated T-REx CHO cells for 3 h and cytokine expression (TNF and IFN-γ) and upregulation of the degranulation marker CD107a were assessed by flow cytometry. Data are representative of five independent experiments

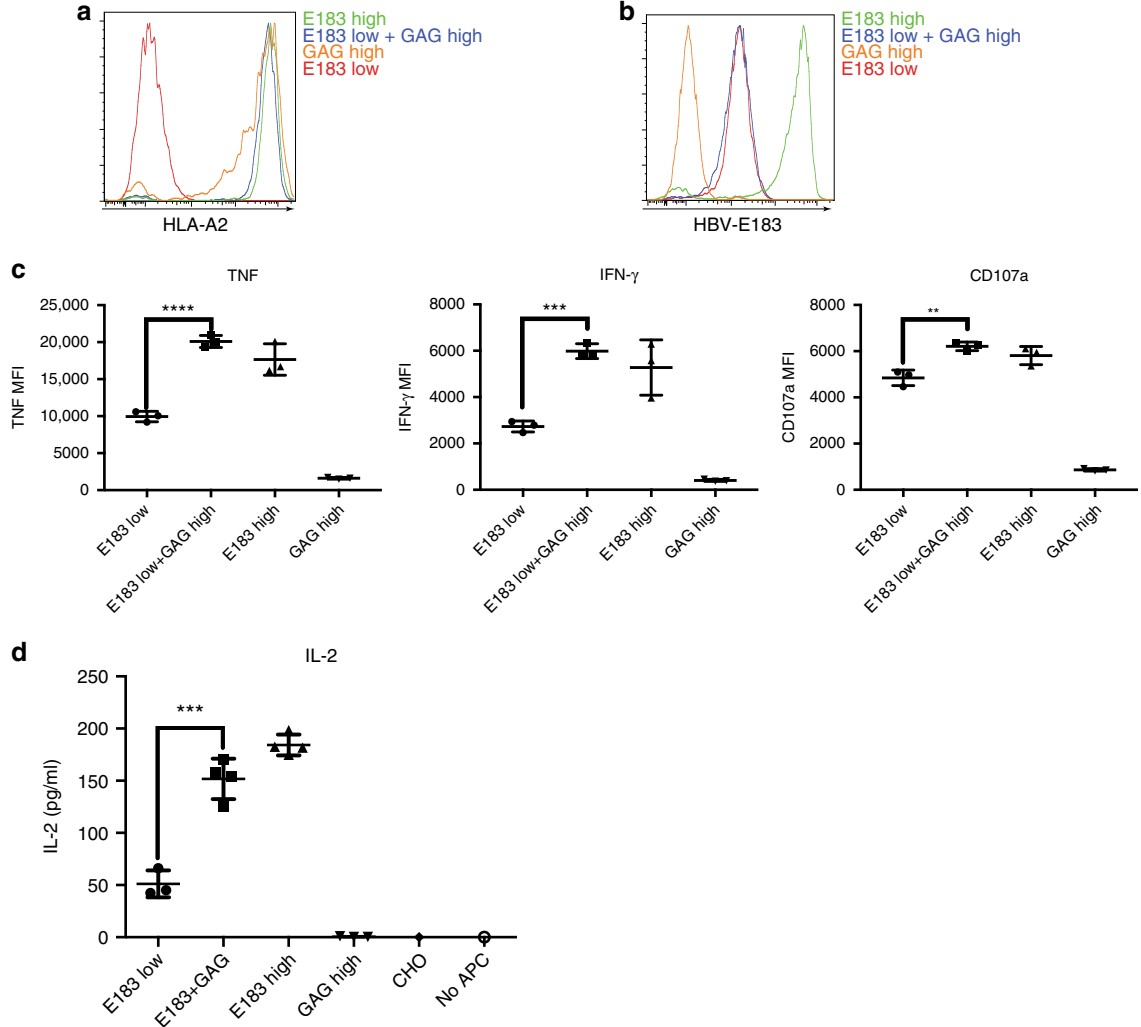

**Fig. 2** Nonstimulatory pMHCI can enhance human CTL response to antigen. **a** Anti-HLA-A2 staining of T-REx CHO clones used in the coagonism experiments. **b** Measurement of the amount of the antigen, E183-HLA-A2 in T-REx CHO clones by staining with TCR-like antibody anti-E183-HLA-A2. **c** Human E183 CTL were cultured with the indicated T-REx CHO clones panel for 3 h and cytokine expression (TNF and IFN-γ) and upregulation of the degranulation marker CD107a were assessed by flow cytometry. **d** Human E183 CTL were cultured with the indicated T-REx CHO clones panel for 24 h, and the amount of IL-2 in the supernatant was assessed by ELISA. Statistical significance was determined by using unpaired two-sided Student's *t* test and shown as mean ± s.d. (*$p < 0.05$, **$p < 0.01$, ***$p < 0.001$). Data are representative of five independent experiments. MFI mean fluorescence level

degranulation marker CD107a, and intracellular expression of cytokines TNF and IFN-γ were analyzed by flow cytometry. The data show that the simultaneous expression of sc GAG-HLA-A2 significantly enhanced the anti-E183 CTL functional responses (Fig. 2c). The same T-REx panel was also cultured with the anti-E183 CTL for 24 h and an IL-2 ELISA was performed. The result showed that high expression of sc GAG-HLA-A2 also enhanced IL-2 secretion (Fig. 2d). In order to confirm coagonism in a different human T-cell system, similar experiments were also performed in TCR-transduced (TCR-Td) human T cells expressing C18-specific TCR or E183-specific TCR[19]. After 3 h stimulation by the respective T-REx CHO cells panel, nonstimulatory sc E183-HLA-A2 or sc GAG-HLA-A2 enhanced C18-specific-TCR-Td T-cell activation in response to antigen sc C18-HLA-A2, and nonstimulatory sc C18-HLA-A2, and sc GAG-HLA-A2 enhanced E183-specific TCR-Td T-cell activation in response to antigen sc E183-HLA-A2 (Supplementary Fig. 2). Therefore, our data show for the first time that the nonstimulatory pMHC can aid in the response of human T cells to antigenic pMHC in the context of physiological interactions between T cell and antigen-presenting cell.

**CD8 binding to coagonist pMHC is required for coagonism.** In the murine immune system, CD8 binding to the coagonist pMHC is required for coagonism. There are several well-studied mutations on HLA-A that can alter the binding affinity of CD8 for HLA class I. The D227K, T228A double mutation abrogates CD8 binding;[20] the A245V mutation reduces CD8-binding affinity fourfold;[21] whereas the Q115E mutation enhances CD8-binding affinity 1.5-fold[22]. First, we tested if these mutations could alter the CD8 binding by introducing the mutation into sc E183-HLA-A2 in pcDNA5/TO vectors. Functional activation data showed that the D227K, T228A mutation on the agonist pMHC complex can effectively abrogate T-cell activation and that the A245V mutation reduced IFN-γ production by 25% (Fig. 3a). To test the requirement of CD8 binding to the coagonist pMHC for coagonism activation enhancement, the three mutations were separately introduced into sc GAG-HLA-A2 (sc GAG-HLA-A2-CD8mut) in pcDNA3-Clover and were then supertransfected into T-REx cells expressing tet-inducible sc E183-HLA-A2. Total HLA-A2 and antigen E183-HLA-A2 expression were assessed by flow cytometry (Fig. 3b, c). These cells were then cocultured with the anti-E183 CTL for 3 h and cytokine production was

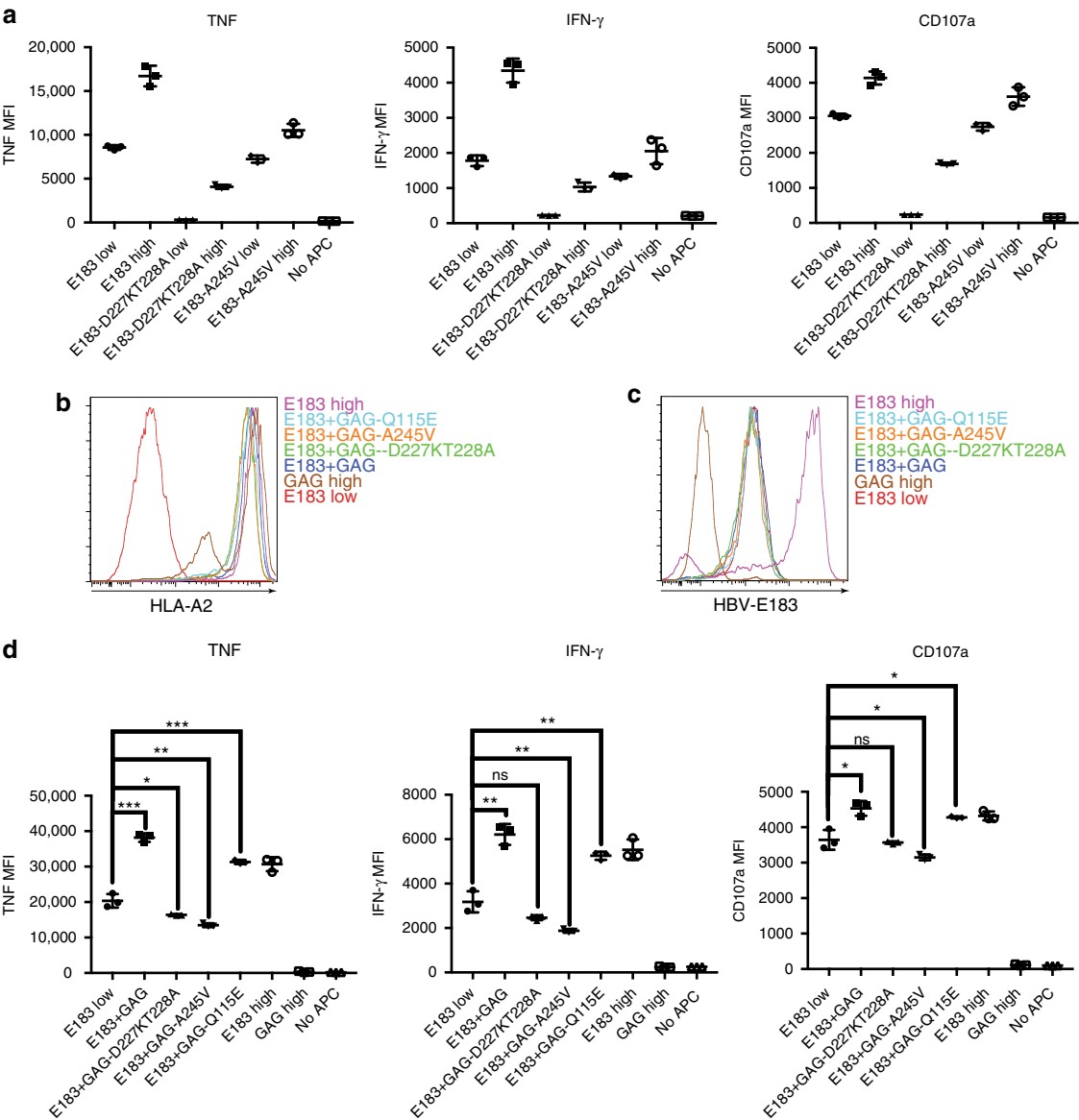

**Fig. 3** CD8 binding to nonstimulatory pMHCI is required for coagonism in human T cells. **a** D227K, T228A, or A245V mutations were introduced in the antigen sc E183-HLA-A2 (sc E183-HLA-A2-CD8mut). Doxycycline-inducible sc E183-HLA-A2-CD8mut constructs were transfected into T-REx CHO cells. Single cell clones were used to stimulate human E183 CTL for 3 h. Cytokine production (TNF and IFN-γ) and upregulation of the degranulation marker CD107a in E183 CTL were assessed by flow cytometry. **b** D227K, T228A, A245V, or Q115E mutations were introduced in the nonstimulatory sc GAG-HLA-A2 (sc GAG-HLA-A2-CD8mut). sc GAG-HLA-A2-CD8mut constructs were transfected into the T-REx CHO clone that could simultaneously express low concentration of the antigen sc E183-HLA-A2 and single cell clones were sorted. Anti-HLA-A2 staining (**b**) or anti-E183-HLA-A2 staining using TCR-like antibody (**c**) was performed on the indicated T-REx CHO clones. **d** Human E183 CTL were cultured with the indicated T-REx CHO cells for 3 h and cytokine production (TNF and IFN-γ) and upregulation of the degranulation marker CD107a were assessed by flow cytometry. Statistical significance was determined by using unpaired two-sided Student's *t* test and shown as mean ± s.d. (ns nonsignificant, *$p < 0.05$, **$p < 0.01$, ***$p < 0.001$, ****$p < 0.0001$). Data are representative of three independent experiments. MFI mean fluorescence level

determined. Compared to wild type HLA-A2 coagonist, the CD8mut variants (D227K, T228A, and A245V) of sc GAG-HLA-A2 could not enhance the anti-E183 CTL response (Fig. 3d). Therefore, CD8 binding to coagonist is required for coagonism in human T-cell activation. The enhanced binding CD8mut-Q115E variant of sc GAG-HLA-A2, however, could not further enhance the coagonism.

**TCR binding to coagonist is not required for coagonism**. TCR recognition of the coagonist pMHC was shown to be important in coagonist enhancement of CD4+ T-cell activation and for the F5 CD8+ T-cell activation[3,8]. However, in OT-I CD8+ T cells several

nonstimulatory pMHC could act as coagonists, demonstrating that the OT-I TCR does not discriminate between coagonist pMHC complexes[4,5,8]. In order to investigate the requirement for TCR recognition of coagonist in human T-cell coagonism, we introduced the K66A mutation into sc E183-HLA-A2 in pcDNA5TO (Supplementary Fig. 3A). This mutation was previously shown to abrogate TCR recognition in a wide range of TCRs restricted by HLA-A2 (Tax$_{11-19}$ and M1$_{58-66}$ peptide)[23–25], but it did not totally abrogate E183 CTL TCR recognition of sc E183-HLA-A2-K66A (Supplementary Fig. 3B). We also introduced the K66A mutation into sc GAG-HLA-A2 in pcDNA3-Clover. The supertransfection of sc GAG-HLA-A2-K66A into the

clone expressing tet-inducible E183-HLA-A2 showed that expression of sc GAG-HLA-A2-K66A was lower than the non-mutant HLA-A2 (Supplementary Fig. 3C). We inferred that the K66A mutation affects peptide binding to the MHC grove and makes the pMHC complex unstable. We also tested the E166K mutation that was used in the mouse coagonism study[8] (Supplementary Fig. 3A). Similar to K66A, the E166K mutation could not abrogate TCR recognition in E183 CTL (Supplementary Fig. 3B). Based on previous research[26], we then tested alternative mutation sites such as R65A, V152E, E154K, Q155K, L156F, and E161V by introducing these mutations into sc E183-HLA-A2 and stimulating anti-E183 CTL using transiently transfected T-REx cells. Of these mutants, only V152E completely abrogated TCR recognition (Supplementary Fig. 4A, B). After selecting and cloning T-REx cells expressing the V152E mutant of sc E183-HLA-A2, we confirmed that V152E abrogated TCR recognition of E183 (Fig. 4a). We also introduced the V152E mutation into GAG-HLA-A2, and demonstrated that this also abrogated GAG-A2-specific TCR-transduced T-cell[27] recognition of the GAG-A2

complex (Supplementary Fig. 5). We then supertransfected sc GAG-HLA-A2 into the clone expressing inducible sc E183-HLA-A2. After antibiotic selection and single cell sorting, we evaluated coagonism. The total HLA-A2 and antigen sc E183-HLA-A2 expression were measured by antibody staining and flow cytometry, showing that the clones expressed similar amounts of HLA-A2 (Fig. 4b) and antigen E183-A2 (Fig. 4c). In coagonism experiments, the sc GAG-HLA-A2-V152E was still able to enhance T-cell activation to repressed E183-HLA-A2 (Fig. 4d). Therefore, TCR binding to the coagonist ligands is not required for coagonism during activation in E183 CTL line.

**CD8 binding to coagonist is sufficient to activate T cells.** The classical model of T-cell activation typically shows the TCR and coreceptor binding to the same antigenic pMHC and initiating TCR signaling (see Fig. 11 of Chapter 7 of Ref. 28)[2,28]. Hoerter et al.[8] showed that TCR binding to nonstimulatory pMHC is required for coagonism when the CD8 binding to antigenic

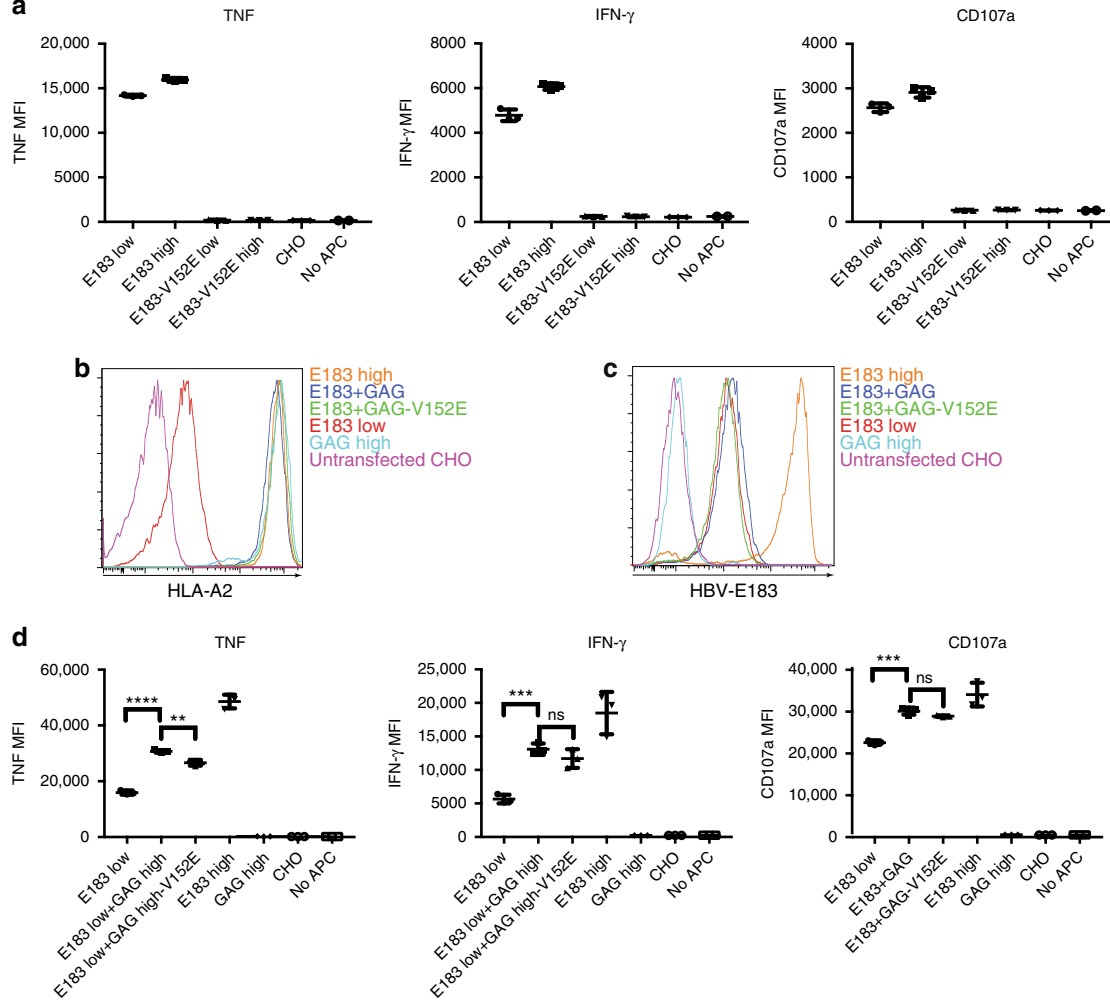

**Fig. 4** TCR binding to nonstimulatory pMHCI is not required for coagonism in human T cells. **a** V152E mutation was introduced into the antigen sc E183-HLA-A2 which was then transfected into T-REx CHO cells and single cell clones were sorted. Indicated T-REx CHO clones panel was cocultured with human E183 CTL for 3 h. Cytokine production (TNF and IFN-γ) and upregulation of the degranulation marker CD107a were assessed by antibody staining and flow cytometry. **b** V152E mutation was introduced into nonstimulatory sc GAG-HLA-A2 which was then transfected into the T-REx CHO clone that could simultaneously express low concentration of the antigen sc E183-HLA-A2. Anti-HLA-A2 (**b**) or anti-E183-HLA-A2 (**c**) staining was performed on the indicated T-REx CHO cells and analyzed by flow cytometry. **d** Human E183 CTL were cocultured with the indicated T-REx CHO clones panel for 3 h. Cytokine production (TNF and IFN-γ) and upregulation of the degranulation marker CD107a were assessed by antibody staining and flow cytometry. Statistical significance was determined by using unpaired two-sided Student's t test and shown as mean ± s.d. (ns nonsignificant, *p < 0.05, **p < 0.01, ***p < 0.001, ****p < 0.0001). Data are representative of three independent experiments. MFI mean fluorescence level

pMHC is abrogated. To evaluate the need for CD8 binding to antigenic pMHC in human T-cell activation, we introduced the D227K, T228A, or A245V mutations in sc E183-HLA-A2. When only these mutant HLA-A2 molecules were expressed on the surface of T-REx cells, D227K, T228A completely abolished T-cell activation, while A245V strongly reduced activation (Fig. 3a), as expected. However, simultaneous expression of sc GAG-HLA-A2 could significantly rescue and enhance T-cell activation (Fig. 5a, b). This demonstrates that CD8 binding only to nonstimulatory (coagonist) pMHC molecules is sufficient to initiate T-cell signaling when the TCR is able to interact with antigenic pMHC. Therefore, T-cell activation does not require CD8 and TCR to bind to the same antigenic pMHC. Thus, nonstimulatory pMHC plays an important role in initiating and amplifying T-cell activation.

**Lack of CD8 and CD3 downregulation by coagonism**. T-cell activation usually induces downregulation of cell surface CD8 and TCR/CD3[29]. We, therefore, tested if the downregulation of cell surface CD8 and CD3 is further increased during coagonism-mediated T-cell activation enhancement. Our data show that the nonstimulatory pMHC did not significantly induce more downregulation of cell surface CD3 or CD8 compared to low amount antigenic pMHC alone, whereas high expression of antigen strongly downregulated cell surface CD8 and CD3 (Fig. 6). This suggests that coagonism does not amplify T-cell activation by triggering, and subsequent endocytosis of, bystander TCR.

**Coagonist pMHC amplify the proximal TCR signaling strength**. The pseudodimer or the preconcentration models of coagonism suggest that the nonstimulatory pMHC can recruit more Lck-associated coreceptor to the immunological synapse[2,3,6]. In order to test if TCR-proximal signaling pathways play an important role in coagonism, we analyzed phosphorylation of molecules downstream of TCR signaling[30]. Firstly, we tested if coagonism could enhance the amount of active Lck (p-Y394 Lck). We found that the amount of active Lck was relatively stable even in the absence of TCR stimulation (Supplementary Figs. 6, 8), in accordance with previous reports[31]. We then tested if phosphorylation of CD3ζ, presumably by active Lck, was increased due to the presence of coagonist pMHC. Compared to the low amount of antigen alone, the coagonists significantly enhanced the phosphorylation of CD3ζ (Fig. 7a, Supplementary Fig. 8). Similarly, coagonist pMHC amplified phosphorylation of Zap70, LAT, PLC-γ1, Erk, and phosphatase SHP-1, suggesting that proximal TCR signaling is enhanced in the presence of coagonist pMHC (Fig. 7b–f, Supplementary Fig. 8).

In order to investigate proximal TCR signaling at the immunological synapse, we analyzed the recruitment of molecules to the synapse by total internal reflection fluorescence microscopy (TIRF). Supported lipid bilayers were used to present pMHC monomers to stimulate human T cells. Phosphorylated CD3ζ, active Lck, and pZap70 were visualized by antibody staining and TIRF imaging. In the negative control ("No pMHC"), proper formation of synapse was not seen. Therefore, the mean intensity of fluorescence could not be quantified for the

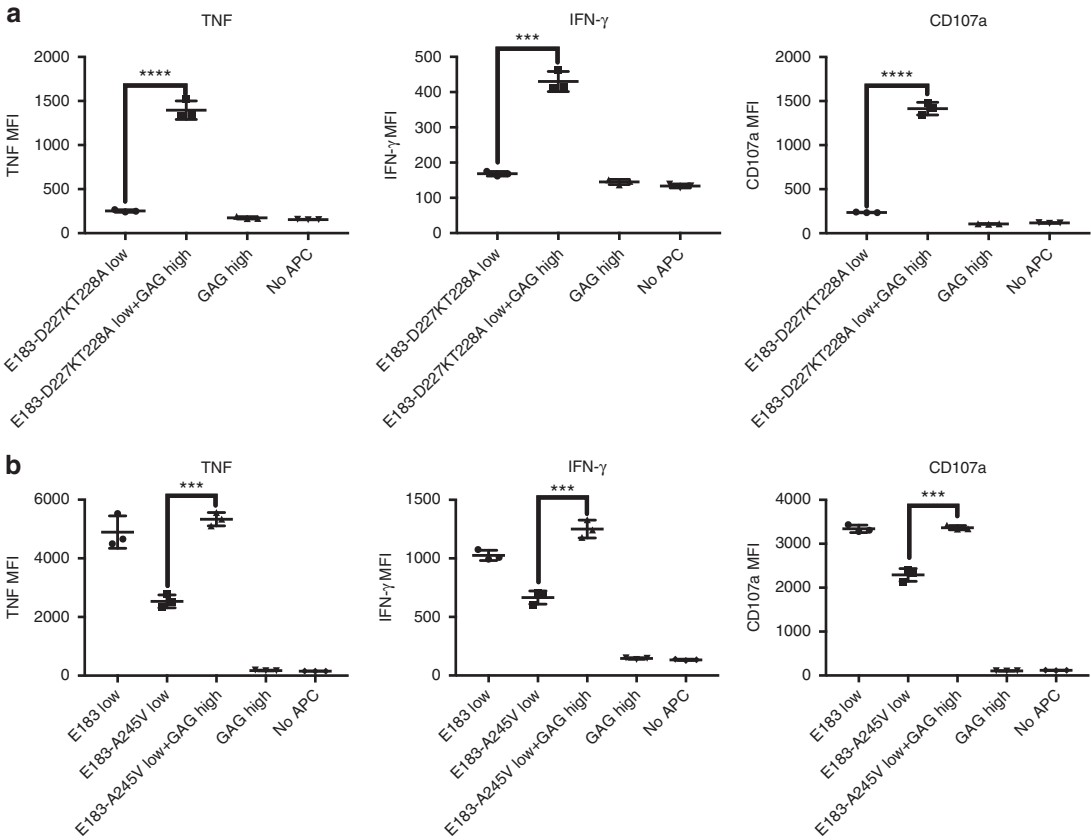

**Fig. 5** Nonstimulatory pMHC can enhance T-cell activation when CD8 binding to antigenic pMHCI is abrogated. sc GAG-HLA-A2 constructs were transfected into the T-REx CHO clone that could express low amount sc E183-HLA-A2 bearing D227K,T228A (**a**) or A245V (**b**) mutation. Single cell clones were sorted. Indicated T-REx CHO clones were co-cultured with human E183 CTL for 3 h. Cytokine production (TNF and IFN-γ) and upregulation of the degranulation marker CD107a were assessed by antibody staining and flow cytometry. Statistical significance was determined by using unpaired two-sided Student's *t* test and shown as mean ± s.d. (ns nonsignificant, *$p < 0.05$, **$p < 0.01$, ***$p < 0.001$, ****$p < 0.0001$). Data are representative of three independent experiments. MFI mean fluorescence level

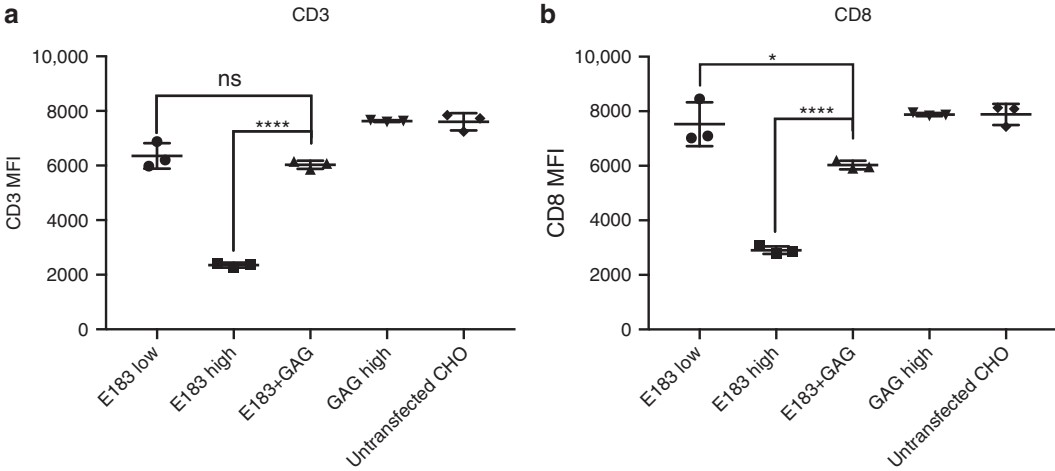

**Fig. 6** CD3 and CD8 are not strongly downregulated in coagonism. Anti-CD3 (**a**) and anti-CD8 (**b**) staining were performed in human CTL activation experiments stimulated by the indicated T-REx CHO clones. CD3 and CD8 amount were then assessed by flow cytometry. Data are representative of three independent experiments. MFI mean fluorescence level

"No pMHC" control. Compared to low concentration of antigenic E183-A2 monomer (10 nM), the simultaneous presentation of high concentration (1 μM) of nonstimulatory GAG-A2 monomer with 10 nM E183-A2 significantly enhanced recruitment of pCD3ζ into the immunological synapse. In contrast, 1 μM GAG-A2 monomer alone recruited very little pCD3ζ into the immunological synapse (Fig. 8a, b). For active Lck, the coagonist GAG-A2 monomer could slightly, but significantly enhance the recruitment of active Lck into the immunological synapse compared to low concentration of antigen alone (Fig. 8c, d). For high concentrations of GAG-A2 monomer alone, synapse formation and recruitment of active Lck to the synapse were seen; this might be because pLck is constitutively present at the cell membrane and can be detected in the TIRF mode (Fig. 8c, d). For pZap70, the coagonist GAG-A2 monomer together with low concentration of antigen significantly enhanced the recruitment of pZap70 into the synapse (Fig. 8e, f). For high concentration of nonstimulatory GAG-A2 monomer alone, there was recruitment of a small amount of pZap70 in the immunological synapse, suggesting that nonstimulatory pMHC alone can induce recruitment of Zap70 (Fig. 8e, f). To study if synapse formation by nonstimulatory GAG-A2 monomer requires intact CD8 binding to the GAG-A2 monomer, D227K/T228A mutation was introduced into the monomer. TIRF imaging results of both pLck and pZap70 showed that the GAG-A2-CD8mut monomer could not form the immunological synapse, establishing that intact CD8 binding to pMHC is required to form the immunological synapse (Supplementary Fig. 7).

## Discussion

Nonstimulatory pMHC complexes have been shown to enhance mouse antigen-specific T-cell responses in multiple experimental systems[3–5,8,32–35], but the molecular mechanism of this phenomenon of coagonism and its relevance to human T-cell responses are poorly understood. We used a human CTL line specific for an HBV-derived peptide and expressed the antigenic pMHC as a single-chain trimer (sc E183-HLA-A2). The HIV-derived peptide GAG was also expressed as a single-chain trimer in order to study coagonism, because it is nonstimulatory for this CTL. We found that sc GAG-HLA-A2 could enhance antigen recognition by the E183-specific CTL, improving effector functions such as the production of cytokines IL-2, TNF, and IFN-γ. This finding is the first demonstration of coagonism in human

T cells responding to antigen presented on APCs. We then used this experimental system to map out the molecular interactions required for coagonism in human CD8+ T cells and to investigate TCR signaling pathways induced by coagonist pMHC recognition.

To enable the study of the molecular interactions required for coagonism in human T-cell activation, different mutations that could abrogate CD8 or TCR binding to MHC were introduced into the coagonist MHC molecules. We found that coagonism in human E183 CTL required CD8 binding to the nonstimulatory pMHC. Interestingly, TCR recognition of the nonstimulatory pMHC was not required. In mouse CD4+ T cells, effective coagonism requires TCR binding to coagonist pMHC since only a small proportion of the nonstimulatory peptides tested could work as coagonists[3]. However, in mouse CD8+ T cells coagonism usually requires CD8 binding to coagonist pMHC, while the TCR-coagonist binding depends on the affinity of CD8 binding to coagonist pMHC[4,5,8]. Since the affinity of human CD8 binding to HLA is usually lower than that of murine CD8 to H-2 molecules, we had predicted that coagonism in human CD8 T cells would require some degree of TCR binding to the nonstimulatory pMHC complex[8]. However, our current study has revealed that at least in human E183-HLA-A2-restricted CTL, coagonism does not require TCR binding to the coagonist nonstimulatory pMHC.

The molecular mechanism underlying pMHC coagonism is a subject of active study though it is still not known how the presence of nonstimulatory pMHC complexes can amplify the antigen-specific activation of T cells. Li et al.[36] suggest that in CD4+ T cells, CD4 controls the spatial localization of Lck. Upon TCR engagement with antigenic pMHC, CD4 can recruit more endogenous pMHC and trigger the TCR engaged with endogenous pMHC. Krogsgaard et al.[3,6] proposed a pseudodimer pMHC model in CD4+ T cells. In this model, CD4 acts as a bridge to connect one antigenic pMHC with one nonstimulatory pMHC to recruit more nonstimulatory pMHC into the synapse. In murine CD8+ T cells, Yachi et al.[4] found that nonstimulatory pMHC alone was able to recruit the coreceptor into the immunological synapse. We suggested a "preconcentration" model in which the nonstimulatory pMHC can enhance the interaction between coreceptor and TCR, resulting in enhanced recruitment of pMHC and coreceptor-Lck into the immunological synapse to accelerate the scanning of different peptides[2]. However, the role of coagonist pMHC recognition in the modulation of downstream signaling pathways is unknown. Critically, it is not known

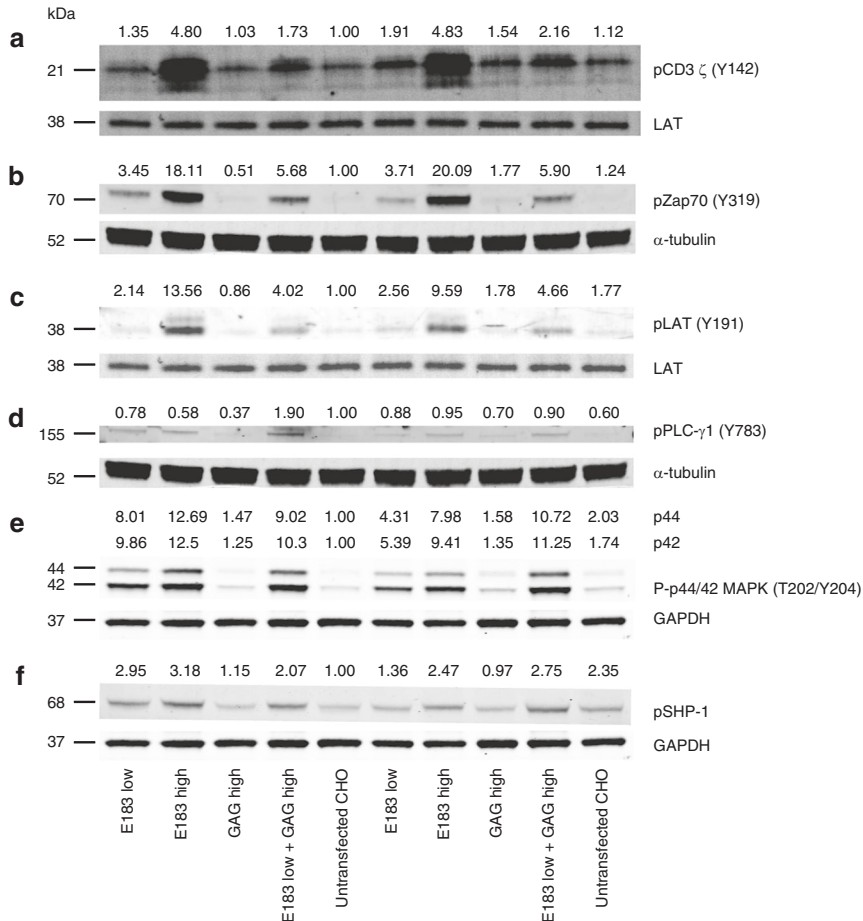

**Fig. 7** Coagonist pMHC enhance human T-cell response to antigens by increasing phosphorylation of molecules involved in proximal TCR signaling pathway. Human E183 CTL were stimulated by T-REx CHO cells panel for 5 or 30 min, lysed and the lysate was separated on NuPAGE Bis-Tris Gel. The filter was blotted with antibodies against pCD3ζ (**a**), pZap70 (**b**), pLAT (**c**), pPLC-γ1 (**d**), p-Erk1/2 (**e**), and pSHP-1 (**f**). The data are representative of three independent experiments. Uncropped western blotting images are presented in Supplementary Fig. 8

if the presence of coagonist pMHC complexes can enhance CD3 ITAM phosphorylation, and whether such an increase is due to an increased phosphorylation of TCR/CD3 complexes bound to antigenic pMHC, or due to an increased phosphorylation of "bystander" TCR/CD3 complexes not engaged by antigenic pMHC. We, therefore, investigated the effect of coagonist pMHC on TCR signaling pathways. We found that coagonists enhanced the phosphorylation of CD3ζ, Zap70, LAT, PLC-γ1, and Erk, suggesting that the TCR signaling strength was amplified by the coagonist. The total amount of active (phosphorylated Y394) Lck was not affected by different conditions of stimulation, consistent with previous reports[31]. We took advantage of TIRF microscopy to study the role of coagonist pMHC complexes in the recruitment of signaling molecules to the immunological synapse. Unlike total active Lck amount detected in the western blotting experiments, the amount of active Lck recruited into immunological synapse was enhanced by the coagonist. pCD3ζ and pZap70 amount were also significantly more enriched in the immunological synapse in the presence of the coagonist.

High concentrations of antigen induced T-cell activation as well as downregulation of TCR and CD8. However, high concentrations of nonstimulatory pMHC could enhance T-cell functional responses to low concentration of antigen, but this did not increase TCR or CD8 downregulation. This shows that the endocytosis of TCR and CD8 is antigen dose-dependent and is not affected by coagonism. Previous work showed that in a dual TCR system, specific pMHC induced the downregulation of

engaged TCR rather than non-engaged TCR[37]. During coagonism, the downregulated TCR are still agonist-engaged, and coagonism may not trigger more bystander TCR to be downregulated. From the western blotting data of pCD3ζ, we found that the phosphorylation of CD3ζ was increased in coagonism but to a lesser extent than during activation by a high concentration of antigen. Previous studies also suggest that the phosphorylation of ITAMs is required for TCR downregulation[38,39]. Therefore, the bystander TCR in coagonism may not be phosphorylated, and is therefore not endocytosed. Otherwise, we would expect to see more phosphorylation of CD3ζ and further TCR downregulation. This could indicate differences in signaling pathways in coagonist-aided activation compared to high concentration antigen-induced activation, with coagonism-induced activation enhancement leading to increase in phosphorylation of TCR/CD3 complexes engaged by the agonist TCR, rather than increase in the number of phosphorylated TCR/CD3 complexes.

In conclusion, we propose an analog amplifier model for the mechanism of coagonism in human CD8[+] T cells. The requirement of CD8 binding to nonstimulatory pMHC in coagonism and the enhanced recruitment of Lck into the immunological synapse suggest a CD8-Lck enrichment into the immunological synapse driven by nonstimulatory pMHC. The increased phosphorylation of CD3ζ, but not of TCR downregulation indicates that only TCRs engaged with agonist were further phosphorylated. Coagonism in human E183 CTL does not require TCR

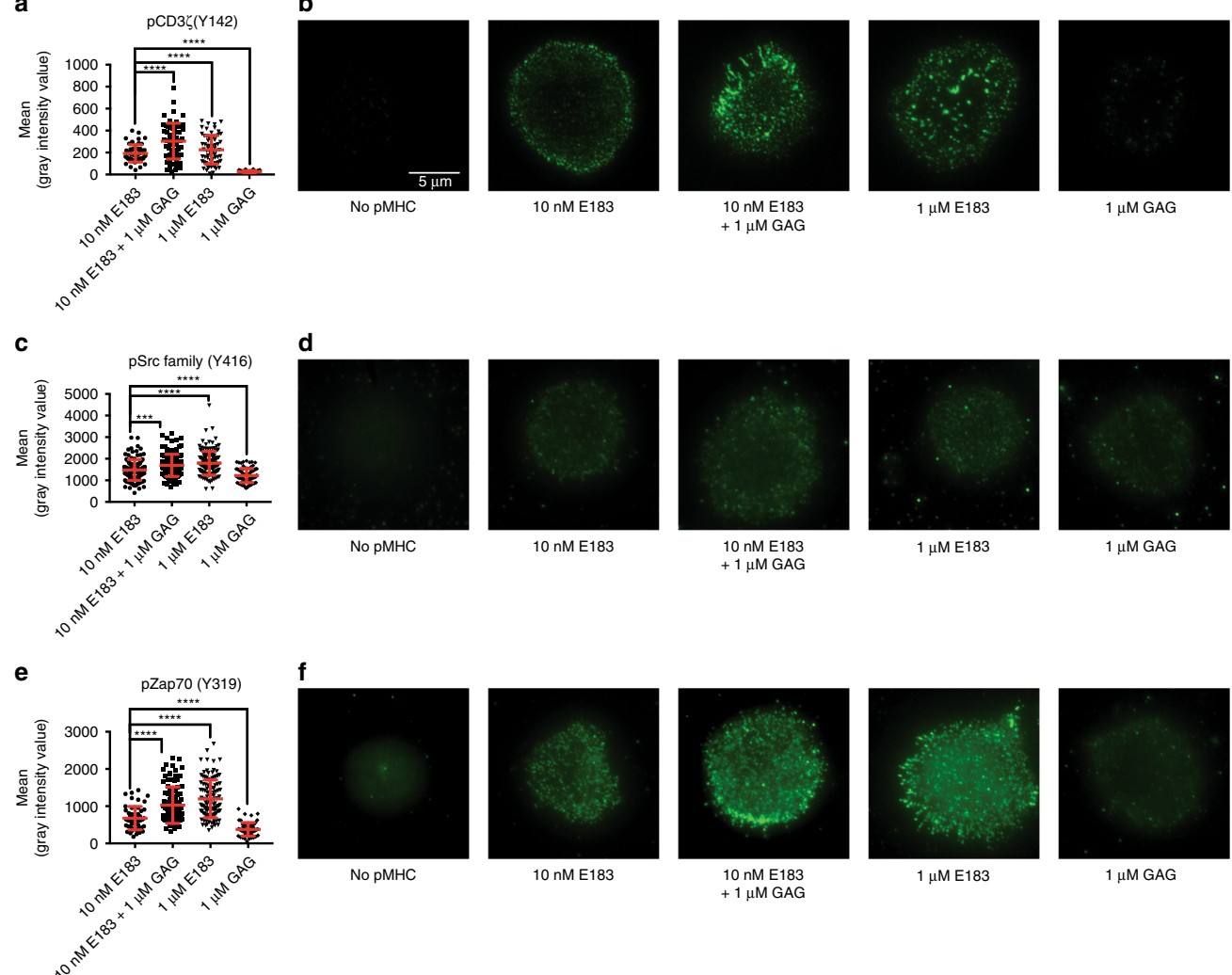

**Fig. 8** Coagonist pMHC enhance the recruitment of pCD3ζ, pLck, and pZap70 into the immunological synapse. Different concentrations of E183-A2 monomer or GAG-A2 monomer were added to the lipid bilayers in a glass chamber slide. Totally, $10^5$ E183 CTL were added into each well of the chamber and stimulated for 5 min. The CTL were fixed, permeabilized and stained with different antibodies to check the recruitment of pCD3ζ (**a, b**), pLck (**c, d**), and pZap70 (**e, f**) into the immunological synapse by TIRF microscopy. Anti-pSrc (Y416) antibody was used to detect active Lck (p-Y394). Statistical significance was determined by using unpaired two-sided Student's $t$ test and shown as mean ± s.d. (ns nonsignificant, $^*p < 0.05$, $^{**}p < 0.01$, $^{***}p < 0.001$, $^{****}p < 0.0001$). Data are representative of three independent experiments

binding to nonstimulatory pMHC, further supporting our hypothesis that bystander TCR may not be triggered in coagonism. Therefore, coagonism likely works by further phosphorylating each antigen-engaged TCR-CD3. For each antigen-engaged T cell, the activation is further enhanced (analog) rather than an all-or-nothing mode (digital). Therefore, we propose that coagonism works in an analog manner by further enhancing the activation of each antigen-engaged T cell. The enhanced phosphorylation of CD3ζ leads to the amplification of downstream signal strength, including the enhanced phosphorylation of Zap70, LAT, PLC-γ1, and Erk. Therefore, coagonism is an amplifier of the entire TCR signaling pathway. The proposed analog amplifier model (Fig. 9) explains how the TCR signaling is initiated and amplified under physiological conditions of APCs presenting limited number of antigens together with large amounts of endogenous nonstimulatory peptide–MHC.

## Methods

**Plasmids**. Single-chain trimer GAG-HLA-A2 was a gift from Dr. Keith Gould (Imperial College London). Peptide mutagenesis: GAG (SLYNTVATL) to E183-91

(FLLTRILTI) or C18-27 (FLPSDFFPSV); D227K-T228A, A245V and Q115E CD8-binding mutations; K66A, R65A, E166K, V152E, E154K, Q155K, L156F, and E161V TCR-binding mutations were all done by using QuickChange Site-Directed Mutagenesis Kit (Stratagene). Agonist single-chain trimer was cloned into tetracycline-inducible pcDNA5/TO (hygromycin resistance, Invitrogen). Nonstimulatory single-chain trimer was cloned into pcDNA3-Clover (geneticin resistance, a gift from Michael Lin [Addgene plasmid # 40259]).

**Antibodies**. Antibodies against HLA-A2 (BB7.2), HLA-ABC (W6/32), CD8a (RPA-T8), TNF (MAb11), and IFN-γ (4 S.B3) were purchased from eBioscience. Anti-CD3 (SK7), CD107a (H4A3), CD247-Alexa Fluor 488 (pY142) (K25-407.69), pCD3ζ (pY142) (K25-407.69), Erk1 (Cat# 610031), and PLC-γ1 (Cat# 610027) were from Becton Dickinson. Anti-pLAT (Tyr191) (Cat# 3584S), P-p44/42 MAPK (T202/Y204) (Cat# 4370L), pSrc family (pY416) (Cat# 6943T), non-pSrc family (pY416) (Cat# 2102S), pSHP-1 (Cat# 8849S), pZap70 (pY319) (Cat# 2717S), Zap70 (Cat# 2709), pPLC-γ1 (pY783) (Cat# 2821S), α-tubulin (Cat# 3873S), GAPDH (Cat# 2118S) antibodies were from Cell Signaling Technologies. Anti-LAT (11B.12) and β-2-microglobulin (2M2) antibodies were from Biolegend. Mouse anti-human HLA-A2-E183-91 and mouse antihuman HLA-A2-C18-27 primary antibodies were produced as described[18]. F(ab')2-goat anti-rabbit IgG (H + L) cross-adsorbed secondary antibody, Alexa Fluor 488 (Cat# A-11070), and goat anti-mouse IgG (H + L) cross-adsorbed secondary antibody, Alexa Fluor 647 (Cat# A-21237) were purchased from Thermo Fisher Scientific.

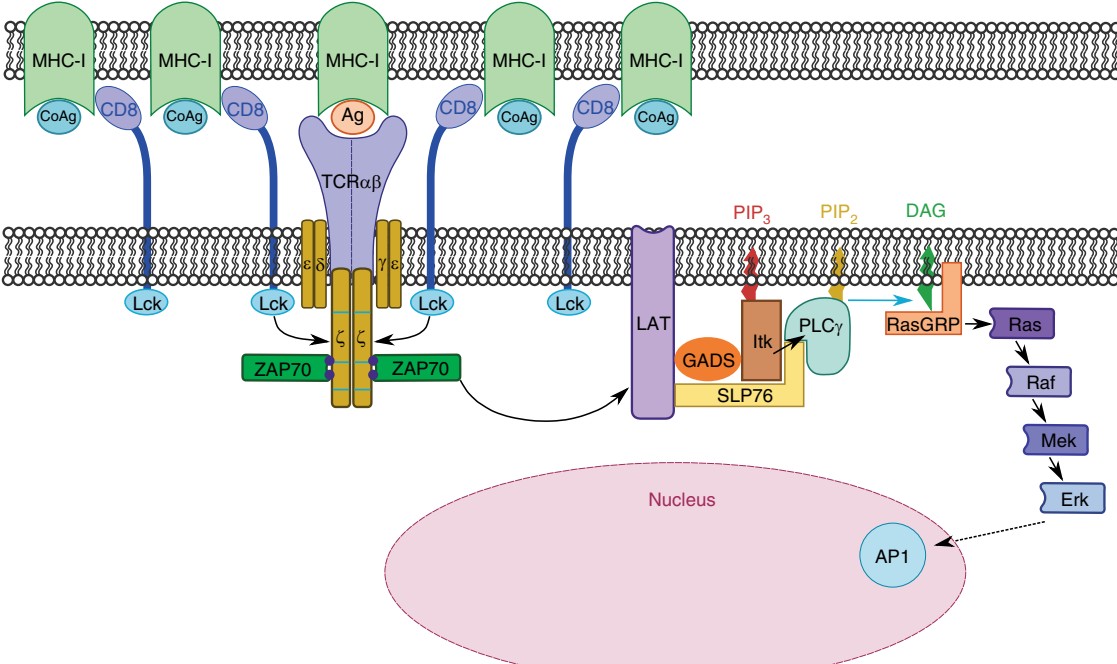

**Fig. 9** Analog amplifier model for the mechanism of coagonism. Nonstimulatory pMHC molecules recruit CD8-Lck into the immunological synapse. Preconcentrated Lck further phosphorylate the ITAMs of TCR-CD3ζ which are engaged with specific antigenic pMHC. Enhanced pCD3ζ provide more docking sites for Zap70 to be phosphorylated. Enhanced pZap70 signal amplifies the downstream TCR signaling pathway including the phosphorylation of Erk. pErk translocates into the nucleus and enhances the effector functional readouts of T-cell activation

**T2-cell assay**. T2 cells were obtained from ATCC. The cells have not been authenticated and not been tested for mycoplasma. The cells were cultured with RPMI-1640 media (Hyclone) supplemented with 10% fetal bovine serum (Hyclone), 2 mM L-glutamine (Gibco), 100 units ml$^{-1}$ penicillin (Gibco), 100 µg ml$^{-1}$ streptomycin (Gibco) and MEM non-essential amino acid (Gibco). $10^6$ T2 cells resuspend in 1 ml media were pulsed with 1 µM E183 peptide (FLLTRILTI) dissolved in DMSO at 37 °C for 1 h. Cells were washed with media to remove extra peptide and stained with anti-HLA-A2 antibody. The MFI of total HLA-A2 were analyzed on flow cytometry.

**CHO cell culture and cloning**. Tetracycline-Regulated Expression (T-REx) CHO cell line was purchased from Invitrogen. The T-REx CHO cells have not been authenticated and not been tested for mycoplasma. The CHO cells were cultured in Ham's F-12 (Gibco) medium with 10% fetal bovine serum (Hyclone), 2 mM L-glutamine (Gibco), 100 units ml$^{-1}$ penicillin (Gibco), 100 µg ml$^{-1}$ streptomycin (Gibco) and 10 µg ml$^{-1}$ blasticidin S HCl (Invitrogen). The blasticidin was used for maintaining the expression of the tetracycline repressor. Transfection of CHO was done by using the polyethylenimine method. After transfection, the cells were selected by the addition of drugs (0.3 mg ml$^{-1}$ hygromycin B from Invitrogen or 1.0 mg ml$^{-1}$ G418 sulfate from Hyclone). High amount agonist expression was induced by adding 60 ng ml$^{-1}$ doxycycline. One day before the human CTL activation assay, $2 \times 10^4$ CHO cells were seeded per well in a 96-well U bottom plate. In all, $2 \times 10^5$ CHO cells were also seeded per well in a 12-well plate for checking the human pMHC expression.

**Culture and restimulation of human CTL**. E183-91 epitope-specific human CTL, C18-91, or E183-91 epitope-specific TCR-transduced human T cells were made as described[40,41]. The E183 CTL have not been authenticated and not been tested for mycoplasma. PBMC were freshly isolated by Ficoll-Paque PLUS (GE Healthcare Life Sciences) according to the manufacturer's protocol and irradiated at 30 Gy. PBMC were resuspended in Aim-V medium with 2% human serum (Sigma-Aldrich) at a concentration of $2 \times 10^6$ per ml. Human CTL were resuspended in the same medium at $10^6$ per ml. Totally, 500 µl of PBMC and 500 µl of human CTL were pooled per well in a 24-well plate. Final concentrations of 20 U ml$^{-1}$ human recombinant IL-2, 10 ng ml$^{-1}$ of IL-7, 10 ng ml$^{-1}$ of IL-15 (R&D Systems) were added to the human CTL culture. Lectin from *Phaseolus vulgaris* (Sigma-Aldrich) was added to the human CTL culture at a concentration of 1.5 µg ml$^{-1}$. PBMC were collected from healthy volunteers under the protocol approved by NUS IRB. Informed consent was obtained from all donors. GAG-A2-specific TCR-transduced T cells were made as described[27].

**T-cell activation assays**. One day before the T-cell activation assays, human CTL were resuspended in Aim-V media containing 2% human serum without the addition of extra cytokines. Human CTL were counted and resuspended at a concentration of $10^6$ per ml, to which the following were added: anti-CD107a antibody (1:100), GolgiPlug Protein transport inhibitor (Brefeldin A, BD Bioscience, 1:1000). The growth media was removed from the culture plate containing CHO cells, and 200 µl of the human CTL mixture was added into each well of the 96-well plate. Technical triplicates were set for the experiments. The plate was incubated at 37 °C, 5% CO$_2$ for 3 h. Surface and intracellular staining were done according to the manufacturer's protocol (BD Cytofix/Cytoperm, BD Perm/Wash, BD Biosciences)

**Flow cytometry**. Flow cytometry experiments were conducted on BD LSRFortessa X-20 (Becton Dickinson). Cell sorting was conducted on Mo-flo XDP (Beckman Coulter, Inc.). Data analysis were performed on FlowJo (TreeStar). FACS gating strategies are included in Supplementary Fig. 9.

**IL-2 ELISA**. Human CTL were co-cultured with CHO cells for 24 h and the supernatant was collected for human IL-2 ELISA assay, which was performed according to the manufacturer's protocol (eBioscience).

**Supported lipid bilayers**. To prepare 0.2 mol% liposomes, 630 µl of 10 mg ml$^{-1}$ 1,2-dioleoyl-*sn*-glycero-3-phosphocholine, 1.76 µl of 10 mg ml$^{-1}$ 1,2-dipalmitoyl-*sn*-glycero-3-phosphoethanolamine-*N*-Cap-Biotinyl (CAP-biotin-PE), and 1.68 µl of 5 mg ml$^{-1}$ 1,2-dioleoyl-*sn*-glycero-3-[(*N*-(5-amino-1-carboxypentyl) iminodiacetic acid) succinyl] (Ni-NTA-DOGS) were mixed in eppendorf tubes. All the lipids were from Avanti Polar Lipids, Inc. Chloroform was evaporated under N$_2$ at 37 °C and the lipid mixture was dried in a Speedvac for 2 h. 2 ml of deoxygenized ddH$_2$O was then added to dissolve the lipids, which were incubated at 4 °C overnight. The next day, the lipids were sonicated until the solution became transparent and filtered through a 0.22 µm filter. This yielded a 4 mM lipid stock that was deoxygenized and stored at 4 °C until reconstitution. Glass 8-well chamber LabTekII chamber slides (Fisher Scientific) were cleaned in 6 M NaOH for 2 h and rinsed with ddH$_2$O. The lipids were diluted 10-folds in PBS, added to clean chamber slides and incubated for 30 min. Excess liposomes were washed away with 12 ml PBS. The bilayers were blocked with 2 mg ml$^{-1}$ BSA for 30 min. This was followed by incubating with 5 µg ml$^{-1}$ streptavidin for 30 min. After washing the excess streptavidin, biotinylated MHCI, recombinant human ICAM1 Protein, hIgG1-Fc.His Tag (Thermo Fisher Scientific) were added to the bilayers for 30 min. After washing, the bilayers were ready to use.

**pLck and pZap70 imaging by TIRF microscopy**. Human E183 CTL were resuspended at $10^6$ per ml and 100 µl of the cells was added to each well containing the lipid bilayer and the specific pMHC monomer. The CTL were stimulated for 5 min at 37 °C, and the reaction was stopped by adding 100 µl of 8% paraformaldehyde. The cells were fixed at room temperature for 12 min. The samples were permeabilized with 0.3% Triton-X 100 for 4 min at room temperature, then blocked with 10% normal goat serum for 1 h. To visualize pLck or pZap70, the sample was labeled with 1:100 of anti-pSrc (Y416) (Cat# 6943 T) or anti-pZap70 (pY319) (Cat# 2717S) (Cell Signaling Technology) for 1 h at room temperature, followed by incubation with Alexa Fluor 488F(ab') 2 fragment of goat anti-rabbit IgG (H + L) at a working concentration of 5 µg ml$^{-1}$ (Invitrogen). TIRF microscopy was performed on an Olympus IX83 inverted microscope fitted with a four-laser TIRF module. Images were acquired using a 100×/1.49 NA oil-immersion lens. Fluorescence excited within the 100-nm evanescent field was recorded with a Hamamatsu ORCA Flash 4.0 camera.

**Western blotting**. $10^5$ T-REx CHO cells were seeded per well in a 24-well plate. The T-REx CHO cell panel used was: E183 low, E183 high, GAG high, E183 + GAG, and untransfected CHO. The human E183 CTL were kept in Aim-V media without any cytokines overnight. After overnight growth, $10^6$ human E183 CTL were added into each well of CHO cells, and incubated at 37 °C, 5% $CO_2$ for 5 min or 30 min. The mixture of CHO cells and E183 CTL were collected and lysed in 120 µl of maltoside lysis buffer. The samples were loaded in a 4–12% Bis-Tris gradient gel (NuPAGE, Invitrogen) and transferred to a PVDF membrane (Immobilon- FL Transfer Membrane, Merck Millipore). The membrane was then blocked using blocking buffer (Odyssey, LI-COR) for 1 h at room temperature. Subsequently, the membrane was probed with different primary antibodies. The secondary antibodies used were IRDye 800CW Goat anti-Mouse IgG2b (Cat# 926–32352, LI-COR) and IRDye 680LT Goat anti-Rabbit (Cat#926-68021). The blotting was quantified by the LI-COR Odyssey infrared imaging system.

**Exclusion criteria and randomization of samples**. No samples were excluded from the analysis. No randomization was used for samples.

**Statistical information and data analysis**. Two-tailed Student $t$ test analysis was performed using GraphPad Prism 7. The data meet the assumptions of the tests. Variance is similar between the compared groups.

**Data availability**. The datasets generated during and/or analyzed during the current study are available from the corresponding author on request.

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

## Acknowledgments

This research was supported by the Singapore Ministry of Health's National Medical Research Council under its CBRG/0064/2014 to N.R.J.G., by CREATE under its R571-002-012-592 to P.A.M., National Research Foundation Investigatorship R571-000-272-281 to P.A.M., and by a Singapore Translational Research (STaR) Investigator Award (NMRC/STaR/013/2012) to A.B. A.K.S. is a Wellcome Senior Investigator. M.L. is funded by a Consolidator Award via the Wellcome Institutional Strategic Support Fund to the Cardiff University College of Biomedical and Life Sciences. We are grateful to Elijah Chen for graphics.

## Author contributions

X.Z. and J.B. designed and conducted experiments; X.Z., J.B., and N.R.J.G. analyzed the data; S.S. aided with imaging experiments and analysis; J.Y. provided technical help; C.T. T., Z.Z.H., E.C.R., G.D., M.L., A.K.S., A.B., and P.A.M. provided reagents; X.Z., J.B., and N.R.J.G. wrote the paper, with input and final approval from all authors.

## Additional information

**Competing interests:** The authors declare no competing interests.

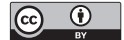

