## [Peer Review File · Nature Communications]

Reviewers' comments:

Reviewer #1 (TCR signaling, T cell biology)(Remarks to the Author):

Previous studies by this and other groups suggest that T cell responses to stimulatory (agonist) peptide/MHC ligands can be enhanced by "co-agonist" peptide/MHC molecules, which are not themselves directly stimulatory. Studies using mouse CD4+ and CD8+ have revealed co-agonism, although initial studies reached contradictory conclusions about whether TCR and/or co-receptor engagement with co-agonists is critical for the effect (and this group proposed that these apparent discrepancies are due to differences in coreceptor binding to MHC molecules and alleles). Little is known about whether responses by human CD8 T cells are also enhanced by co-agonists and – if so – whether such responses would involve TCR and/or CD8 engagement with the coagonists. Since coagonism could present a mechanism to enhance weak TCR stimulation (e.g. due to low Class I MHC expression in virally infected cells), these studies are significant.

The current study shows that co-agonism occurs for human CTL specific for an HBV epitope, that the co-agonist ligand needs to bind CD8 but (probably) not the TCR, and that the results of co-agonism are to enhance early TCR signaling events and immunological synapse formation, leading to improved functional responses. These data add to our understanding of the relevance and mechanism of co-agonism in T cell immune responses, potentially offering new opportunities to enhance suboptimal T cell activation for therapeutic goals. There are a few concerns, however.

1) The authors' data on the lack of TCR recognition of co-agonists is critical for establishing their model (and distinguishing their findings from previous hypotheses about the foundation of co-agonism). However, the approach used is not convincing. The authors develop a panoply of HLA-A2 a-helix mutations to identify ones that compromise the response of the HBV specific CTL, and find that only V152E (introduced into the agonist E183/HLA-A2 ligand) completely blocks stimulation. They then introduce this same mutation into the co-agonist ligand (HIV GAG/HLA-A2), and show that this V152E mutant still serves as a co-agonist.

However, the authors do not show that this mutation in GAG-A2 affects stimulation of CTL specific for the GAG-A2 complex, and it is quite feasible that, with this particular peptide, this residue is not critical for TCR recognition of the MHC component. Furthermore, as shown in Ref 31, critical MHC contact residues may differ between T cell clones (even ones specific for the same antigen). In keeping with this, the authors own data show that only the V152E mutation dramatically impaired the response of the HBV-specific clone, while Ref 31 showed E154K and E161V were at least as potent (for the flu-specific CTL analyzed in that report).

At the very minimum, the authors need to show that the responses of CTL clones (ideally, several) specific for GAG-A2 are compromised by the V152E mutation. Even then, it seems possible that (weak) TCR recognition of GAG-A2 by the HBV-specific clone could occur in a V152E-independent way. One solution may be to make a combination HLA-A2 mutant, carrying as many of the targeted substitutions as possible – e.g. mutations of V152, E154, Q155, L156, E161 AND E166. It is possible, of course, that this particular composite mutant will not be expressed, but something along these lines may make a more compelling case that the mutant Class I molecule is extremely unlikely to interact with any (normal) HLA-A2 specific TCR.

2) The protracted discussion of the phosphorylated Src-family kinase that may be Fyn and/or an alternatively phosphorylated form of Lck is confusing, unresolved and, to this reviewer, unnecessary for the main thrust of the study. The authors do not help by repeatedly labeling blots as "Lck (pY394)" when they then go to pains to explain that the antibody used is cross-reactive between Src-family kinases, and may not actually be detecting Lck in this context.

Without more clear resolution of this issue, this component of the results seems like a frustrating distraction, and detracts from the clearer (and, in this reviewer's opinion, equally important) findings about co-agonist enhanced phosphorylation of CD3zeta, ZAP70, LAT, PLCgamma-1, ERK1/2 and SHP-1. Perhaps the Lck data could either be removed or placed entirely in the Supplement?

3) The data on weak induction of synapse formation by the co-agonist (GAG-A2) alone, in a CD8-binding-dependent way is shown in Fig. 8 and Supp. Fig. 5, but these two figures show very different signal intensities for synapse formation in response to WT GAG-A2, making it hard to compare the figures. Presumably, the GAG-A2-CD8mut gave a signal which was similar to the no pMHC control – this should be included in the compiled data in Fig. 8A, C and E.

Reviewer #2 (TCR, tumor, crossreactivity)(Remarks to the Author):

This solid piece of work confirms and extends previous studies on co-agonist peptide-MHC undertaken by this group in the mouse that published in the *Journal of Experimental Medicine* 210:1807. The study uses the same methodology as used in the JEM paper, with the only major difference being that the former study examined transgenic murine T cells (OT-1 and F5) whereas the current study uses an HLA-A2 restricted human T cell clone specific for an epitope from the HBV virus.

The article is well written and easy to follow. Previous studies by this group and others over a decade ago have demonstrated that non-stimulatory pMHC augment responses to antigenic pMHC. This effect has been termed “co-agonism”. All the work to date on co-agonist pMHC has exclusively examined murine T cells. Zhao and colleagues now extend these studies to examine human T cells (or more specifically a single HBV-specific human T cell clone). Like the preceding JEM study, antigen presenting cells were constructed from the xenogenic Chinese hamster ovary (CHO) cell line via transduction with single chain peptide-HLA-A2 constructs. The rationale behind the use of CHO cells was to avoid presentation of self peptides via HLA-A2. The ability of various ligands on hamster cells to bind relevant receptors on human T cells and how this might impact on T-cell activation remains unknown. Nevertheless, as in the mouse system, the data clearly show that a CHO cell expressing an agonist scHLA-A2 construct can activate a human T cell clone specific for the FLLTRILT1 HBV peptide. Super-transfection with a scHLA-A2-SLYNTVATL construct augmented recognition of scHLA-A2-FLLTRILT1. Elegant use of mutations in HLA-A2 that are known to abrogate, reduce or enhance binding of the CD8 glycoprotein enabled assessment of the role of this co-receptor in T cell activation. Abrogation of CD8 binding in the agonist construct showed that T cell activation was completely CD8 dependent. The co-agonist affect disappeared when CD8 binding was abrogated in the co-agonist ligand indicating that CD8 binding is required for co-agonist activity. Other experiments using mutant scHLA show that scHLA-A2-SLYNTVATL can rescue recognition of CD8-null scHLA-A2-FLLTRILT1 and indicate that CD8 binding to non-stimulatory pMHC is sufficient to rescue recognition of agonist ligands that lack the capacity for engagement of CD8. Curiously, T cell activation under co-agonist conditions did not result in substantial downregulation of CD3 (TCR) or CD8. Examination of signal transduction showed that the presence of a co-agonist ligand amplified the TCR-mediated signal transduction cascade.

In summary, this study includes a lot of very nice data that must represent many man hours at the bench and the conclusions appear to be justified.

WHAT FURTHER EVIDENCE WOULD STRENGTHEN THE CONCLUSIONS? The results seem to be based on a single HBV-specific T-cell clone/TCR. It would be reassuring to see reciprocal experiments in the same system using SLYNTVATL-specific T-cells. In this case the EBV peptide would act as the co-agonist so these experiments would not require manufacture of additional CHO antigen presenting cells. Some experiments with a SLYNTVATL T cell clone/TCR would improve the study and strengthen the conclusions.

MINOR CONCERNS: Many groups, and a great many studies, have made extensive use of T2 peptide binding assays to assess the ability of exogenous peptide to stabilize HLA-A2 by flow cytometry. Supplementary Figure 1 does not fit these data. The reasons for this might be due to the peptide used or the concentration it was used at. It is impossible to tell what was done to generate this figure as neither the peptide concentration, its sequence or the length of time it was applied for are given in the legend to Supplementary Figure 1 or in the materials and methods. In my experience, the T2 cells HLA-A2 peptide binding stabilization requires peptides with good P2 and C-terminal primary HLA-A2 anchors and peptide concentrations of 10 uM or higher (preferable at least 100 uM). The study would be improved if the authors show a range of concentrations 1-100 uM for a peptide that is known to bind to HLA-A2 well (e.g. influenza M1 58-66). While the data in Supplementary Figure 1 are poor, and do not conform to previous results from several other groups, this deficiency does not really impact on the conclusions of the current study.

Three mutations that affect CD8 binding were used. The 227/8KA mutant abrogated T-cell activation while the A245V mutation that reduces CD8 binding reduced T cell activation. The effects of the third Q115E mutation, which is said enhance CD8 binding, in the agonist ligand was not described. Is activation by this mutant similar to, or greater than, that with wild type HLA-A2? If the authors made the construct then I assume that they must have the relevant data so I am surprised that it was not included. Was there a reason for not including Q115E data for the agonist ligand?

Response to reviewer #1

- 1) The authors' data on the lack of TCR recognition of co-agonists is critical for establishing their model (and distinguishing their findings from previous hypotheses about the foundation of co-agonism). However, the approach used is not convincing. The authors develop a panoply of HLA-A2 a-helix mutations to identify ones that compromise the response of the HBV specific CTL, and find that only V152E (introduced into the agonist E183/HLA-A2 ligand) completely blocks stimulation. They then introduce this same mutation into the co-agonist ligand (HIV GAG/HLA-A2), and show that this V152E mutant still serves as a co-agonist.

However, the authors do not show that this mutation in GAG-A2 affects stimulation of CTL specific for the GAG-A2 complex, and it is quite feasible that, with this particular peptide, this residue is not critical for TCR recognition of the MHC component. Furthermore, as shown in Ref 31, critical MHC contact residues may differ between T cell clones (even ones specific for the same antigen). In keeping with this, the authors own data show that only the V152E mutation dramatically impaired the response of the HBV-specific clone, while Ref 31 showed E154K and E161V were at least as potent (for the flu-specific CTL analyzed in that report).

At the very minimum, the authors need to show that the responses of CTL clones (ideally, several) specific for GAG-A2 are compromised by the V152E mutation. Even then, it seems possible that (weak) TCR recognition of GAG-A2 by the HBV-specific clone could occur in a V152E-independent way. One solution may be to make a combination HLA-A2 mutant, carrying as many of the targeted substitutions as possible – e.g. mutations of V152, E154, Q155, L156, E161 AND E166. It is possible, of course, that this particular composite mutant will not be expressed, but something along these lines may make a more compelling case that the mutant Class I molecule is extremely unlikely to interact with any (normal) HLA-A2 specific TCR.

Reply:

We are grateful to the reviewer for commenting on this important issue. The functional effects of any MHC mutation are predicted to be highly dependent on the TCR tested. Therefore, it is not surprising that only V152 mutation abrogated TCR binding in the E183-specific clone used, even if other HLA mutants were previously reported to reduce binding of other TCRs. Since our data show that V152 mutation completely abolishes activation of the E183-specific clone in response to the strong affinity agonistic E183-HLA, we find it extremely unlikely that this mutation could allow any binding of our E183-specific TCR to the subthreshold affinity HLA-GAG. However, to further validate that the V152 mutation can abrogate TCR binding in the context of recognition of GAG-A2 complexes, we followed the reviewer's suggestion and used GAG-A2 complex-specific CTL to test the requirement of V152 residue for the CTL recognition of GAG-A2 complex. We observed that the V152E mutation abrogated the GAG-A2-specific TCR recognition of GAG-A2 complex (Figure 1). Therefore, the V152E mutation is also important in the recognition of GAG-A2 complex by GAG-A2-specific TCR.

Figure 1. V152E mutation abrogates specific TCR recognition of GAG-A2 complex. GAG-A2-specific T cells were cocultured with the indicated CHO cells panel for 3 h, and the production of cytokine (TNF- α and IFN- γ) and upregulation of CD107a were measured.

Unfortunately, the other proposed solution of making a combination HLA-A2 mutant will be very time-consuming in our experimental system, as it would require generation of multiple new CHO cell lines. Therefore, we didn't consider this solution in practice.

- 2) The protracted discussion of the phosphorylated Src-family kinase that may be Fyn and/or an alternatively phosphorylated form of Lck is confusing, unresolved and, to this reviewer, unnecessary for the main thrust of the study. The authors do not help by repeatedly labeling blots as "Lck (pY394)" when they then go to pains to explain that the antibody used is cross-reactive between Src-family kinases, and may not actually be detecting Lck in this context.

Without more clear resolution of this issue, this component of the results seems like a frustrating distraction, and detracts from the clearer (and, in this reviewer's opinion, equally important) findings about co-agonist enhanced phosphorylation of CD3zeta, ZAP70, LAT, PLCgamma-1, ERK1/2 and SHP-1. Perhaps the Lck data could either be removed or placed entirely in the Supplement?

Reply:

The Lck data have been placed in the Supplementary Figure 5. Relevant results and discussion have also been revised accordingly.

- 3) The data on weak induction of synapse formation by the co-agonist (GAG-A2) alone, in a CD8-binding-dependent way is shown in Fig. 8 and Supp. Fig. 5, but these two figures show very different signal intensities for synapse formation in response to WT GAG-A2, making it hard to compare the figures. Presumably, the GAG-A2-CD8mut gave a signal which was similar to the no pMHC control – this should be included in the compiled data in Fig. 8A, C and E.

Reply:

The data from Figure 8 and Supplementary Figure 5 were from two independent experiments, so we did not compile them. The reason for the apparent different signal intensities is due to use of different scaling to produce the images. In order to address the reviewer's concern, we revised the supplementary Figure 5, which used the exactly same scaling as Figure 8, so that the signal intensities for synapse formation in response to GAG-A2 are similar between Figure 8 and supplementary Figure 5 (now supplementary Figure 6 in revised version). It must be noted that the choice of scaling does not alter the intensity values, either for raw or background-corrected intensities.

Response letter to reviewer #2

- 4) WHAT FURTHER EVIDENCE WOULD STRENGTHEN THE CONCLUSIONS? The results seem to be based on a single HBV-specific T-cell clone/TCR. It would be reassuring to see reciprocal experiments in the same system using SLYNTVATL-specific T-cells. In this case the EBV peptide would act as the co-agonist so these experiments would not require manufacture of additional CHO antigen presenting cells. Some experiments with a SLYNTVATL T cell clone/TCR would improve the study and strengthen the conclusions.

Reply:

The E183 low + GAG high CHO cell clone was made by using tetracycline-inducible E183-A2 construct and constitutively-expressed GAG-A2 complex. Therefore, we actually do not have a clone that expresses low amount of GAG-A2 and high amount of E183-A2 at the same time. Moreover, it will be very time-consuming to generate such cell line. We therefore adopted an alternative approach to address reviewer's concern by using TCR-transduced T cells.

Two different kinds of TCR-transduced T cells were used: one kind expresses HBV epitope C18-specific TCR (C18-TCR-Td), and another expresses HBV epitope E183-specific TCR (E183-TCR-Td). We observed that for C18-TCR-Td T cells, non-stimulatory peptide E183 and GAG could enhance the T cell response to low amount of antigen C18. In E183-TCR-Td T cells, non-stimulatory peptide C18 and GAG could enhance the T cell response to low amount of antigen E183 (Supplementary Figure 2). Therefore, this data shows coagonism occurs in more than one kind of human T cells.

5) MINOR CONCERNS: Many groups, and a great many studies, have made extensive use of T2 peptide binding assays to assess the ability of exogenous peptide to stabilize HLA-A2 by flow cytometry. Supplementary Figure 1 does not fit these data. The reasons for this might be due to the peptide used or the concentration it was used at. It is impossible to tell what was done to generate this figure as neither the peptide concentration, its sequence or the length of time it was applied for are given in the legend to Supplementary Figure 1 or in the materials and methods. In my experience, the T2 cells HLA-A2 peptide binding stabilization requires peptides with good P2 and C-terminal primary HLA-A2 anchors and peptide concentrations of 10 μ M or higher (preferable at least 100 μ M). The study would be improved if the authors show a range of concentrations 1-100 μ M for a peptide that is known to bind to HLA-A2 well (e.g. influenza M1 58-66). While the data in Supplementary Figure 1 are poor, and do not conform to previous results from several other groups, this deficiency does not really impact on the conclusions of the current study.

Reply: We have extensively used the mouse equivalent of T2 cells – the TAP-deficient RMA-S cells – to study coagonism during murine T cell activation^{1,2}. We therefore decided to include the T2 data showing that this system does not seem to be suitable for work on coagonism in human T cells to save time and effort of any labs that may want to follow up the findings of our study. It must be noted that the requirements for our experiments are not the same as in classic peptide binding stabilisation assays.

We think the main concern is that we want almost no endogenous peptide presented by the cells, just like the mouse Tap-deficient cell line RMA-S used in the mouse coagonism study. The peptide used for this figure was E183 (FLLTRILT) and the concentration was 1 μ M. This information has been added to the relevant figure legend. The peptide stock we used was at 1 mM dissolved in DMSO. We think a

concentration of 10 μ M or 100 μ M peptide in this system is too high due to the DMSO toxicity.

6) Three mutations that affect CD8 binding were used. The 227/8KA mutant abrogated T-cell activation while the A245V mutation that reduces CD8 binding reduced T cell activation. The effects of the third Q115E mutation, which is said enhance CD8 binding, in the agonist ligand was not described. Is activation by this mutant similar to, or greater than, that with wild type HLA-A2? If the authors made the construct then I assume that they must have the relevant data so I am surprised that it was not included. Was there a reason for not including Q115E data for the agonist ligand?

Reply: The truth is that we did not make the agonist-Q115E constructs. In our coagonism experiments, we showed that the Q115E mutation on coagonist couldn't further enhance the coagonism-aided activation. Therefore, we decided not to investigate the effects of Q115E mutation any further, as we do not think this will add any new understanding of the molecular mechanism of coagonism during T cell activation.

Reference

1. Yachi, P. P., Ampudia, J., Gascoigne, N. R. J. & Zal, T. Nonstimulatory peptides contribute to antigen-induced CD8-T cell receptor interaction at the immunological synapse. *Nat. Immunol.* **6**, 785–792 (2005).
2. Hoerter, J. A. H. *et al.* Coreceptor affinity for MHC defines peptide specificity requirements for TCR interaction with coagonist peptide-MHC. *J. Exp. Med.* **210**, 1807–1821 (2013).

REVIEWERS' COMMENTS:

Reviewer #1 (Remarks to the Author):

The authors have addressed previous concerns but it is surprising that they did not include the data presented for reviewers (Fig 1 in the rebuttal) in the manuscript. This should be introduced into the Supplementary data (e.g. SFig. 4) and briefly discussed the text as an important control. The point of raising this issue, after all, was not to satisfy this reviewer's personal curiosity but to strengthen the interpretation of the data for all readers.

Reviewer #2 (Remarks to the Author):

I'm satisfied with the authors responses to the previous concerns.

Response to issues raised by referee

REVIEWERS' COMMENTS:

Reviewer #1 (Remarks to the Author):

The authors have addressed previous concerns but it is surprising that they did not include the data presented for reviewers (Fig 1 in the rebuttal) in the manuscript. This should be introduced into the Supplementary data (e.g. SFig. 4) and briefly discussed the text as an important control. The point of raising this issue, after all, was not to satisfy this reviewer's personal curiosity but to strengthen the interpretation of the data for all readers.

Response: We have included the data in the supplementary data as supplementary Figure 5. (This was done in the previous submission).

Reviewer #2 (Remarks to the Author):

I'm satisfied with the authors responses to the previous concerns.

Response: Thanks!